# VisualSync: Multi-Camera Synchronization via Cross-View Object Motion

**Shaowei Liu**[1*] **David Yifan Yao**[1*] **Saurabh Gupta**[1†] **Shenlong Wang**[1†]

[1]University of Illinois Urbana-Champaign

https://stevenlsw.github.io/visualsync

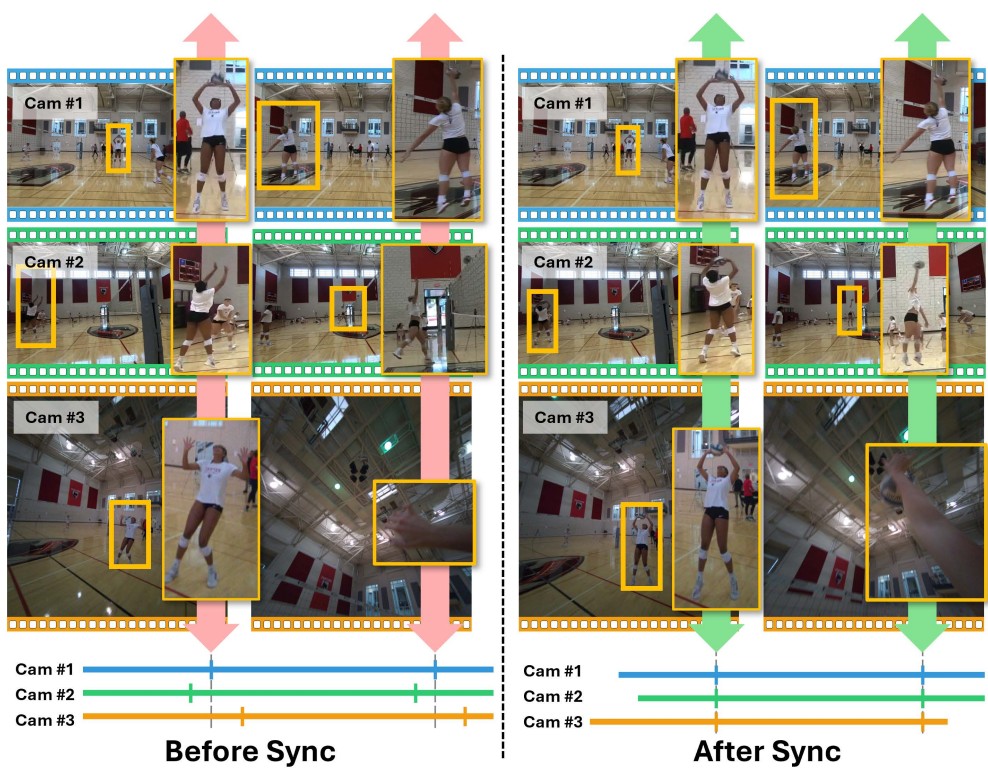

**Figure 1: VisualSync Overview.** Given multiple unsynchronized videos capturing the same dynamic scene from different viewpoints, VisualSync recovers globally time-aligned video streams by estimating temporal offsets between views. For example, in the volleyball scene, before synchronization the player's motion is misaligned across videos; afterwards, a given timestamp in all three streams corresponds to the same moment.

## Abstract

Today, people can easily record memorable moments, ranging from concerts, sports events, lectures, family gatherings, and birthday parties with multiple consumer cameras. However, synchronizing these cross-camera streams remains challenging. Existing methods assume controlled settings, specific targets, manual correction, or costly hardware. We present VisualSync, an optimization framework based on multi-view dynamics that aligns unposed, unsynchronized videos at millisecond accuracy. Our key insight is that any moving 3D point, when co-visible in two cameras, obeys epipolar constraints once properly synchronized. To exploit this,

---

* Equal contribution; † Equal advising.

39th Conference on Neural Information Processing Systems (NeurIPS 2025).

VisualSync leverages off-the-shelf 3D reconstruction, feature matching, and dense tracking to extract tracklets, relative poses, and cross-view correspondences. It then jointly minimizes the epipolar error to estimate each camera's time offset. Experiments on four diverse, challenging datasets show that VisualSync outperforms baseline methods, achieving an average synchronization error below 130 ms.

# 1 Introduction

Recording dynamic scenes from multiple viewpoints has become increasingly common in everyday life. From concerts and sports events to lectures and birthday parties, people often capture the same moment using different handheld devices. These multi-view recordings present a rich opportunity to reconstruct scenes in 4D, enable bullet-time effects, or enhance the capabilities of existing vision models. However, these videos are typically captured independently, without synchronization or known camera poses, making it difficult to align and fuse them coherently.

Existing synchronization methods rely on controlled environments, manual annotations, specific patterns (*e.g.* human pose), audio signals (*e.g.* flashes or claps), or expensive hardware setups such as time-coded devices, none of which are available in casually captured videos. In this work, we design a versatile and robust algorithm for synchronizing videos without requiring specialized capture or making assumptions about the scene content. Our key insight is that, at the correct synchronization, the scene is static and thus the epipolar relationship (*i.e.* $x'^{T}Fx = 0$ for correspondent points $x$ and $x'$ in two views) must hold true for all correspondences, whether on static or dynamic objects [19]. See Fig. 2 for an illustration.

While the insight follows directly from first principles and has been used in past attempts on this problem [3, 34, 60, 41, 31, 55, 16], building a practical and robust system that works on videos in the wild is challenging. We need a reliable estimate for the fundamental matrix between camera pairs, dynamic objects are a priori unknown and are generally small and blurry, and not all views may have an overlap. Our key contribution is to leverage recent advances in computer vision, specifically dense tracking, cross-view correspondences, and robust structure-from-motion, to build a robust and versatile system that can reliably synchronize challenging videos.

Specifically, we formulate a joint energy function that measures the violations to the epipolar constraints between correspondences between videos and adopt a three-stage optimization procedure. In Stage 0, we use VGGT [56] to estimate fundamental matrices between each video pair, use MAST3R [30] to establish correspondences across videos, and use CoTracker3 [24, 23] to establish dense tracks within each video. This gives us access to quantities (correspondences and fundamental matrices) necessary to evaluate the joint energy. Optimizing this joint energy directly is challenging. Therefore, in Stage 1, we decompose this energy into pairwise energy terms and estimate the best temporal alignment between each video pair via a brute force search. In Stage 2, we synchronize the temporal offsets across all video pairs to assign a globally consistent temporal offset to each video.

We validate our approach on diverse datasets and show strong performance across different scenes, motions, and camera setups, and achieve high-precision synchronization even under severe viewpoints. Specifically, we outperform SyncNerf [26], a recent method for this task by radiance field optimization, and adaptations of two recent methods Uni4D [62] and MAST3R [30]. These results demonstrate the robustness and generality of our approach and open the door to scalable, unconstrained multi-view 4D scene understanding.

# 2 Related work

**Tracking and Correspondence.** Establishing reliable correspondences across time and views is fundamental for synchronizing multi-view videos [15, 4, 6, 12, 47, 5, 51]. Recent models like CoTracker [24, 23, 18, 13, 14] track points densely over time, offering strong temporal coherence. However, they do not model spatial correspondences across different viewpoints. On the other hand, MASt3R [30, 58] focuses on spatial matching and stereo reconstruction, providing dense cross-view correspondences, but it does not handle temporal dynamics, especially in moving scenes. Our method bridges this gap by constructing spatio-temporal cross-view correspondences, integrating both temporal tracking and spatial matching to enable accurate synchronization in dynamic, multi-view video settings.

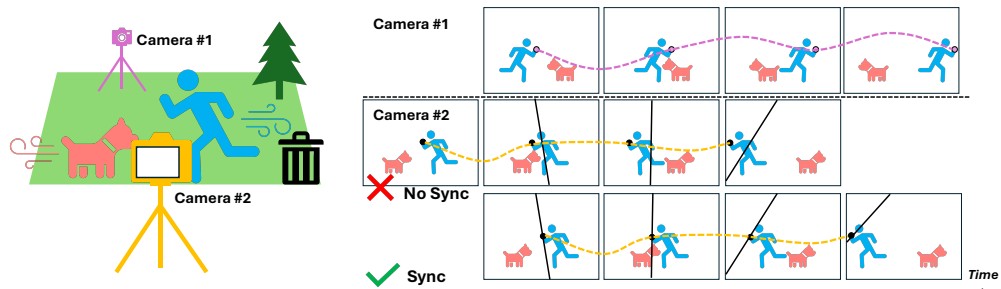

**Figure 2: Epipolar-geometry cue for video sync:** When cameras are time-aligned, keypoint tracks align with epipolar lines (bottom); misalignment causes deviations (middle). Minimizing these deviations across tracklets recovers the correct time offset.

**Multi-View Structure-from-Motion.** Structure-from-Motion (SfM) techniques [2, 58, 48, 52, 38, 39, 50, 61], such as COLMAP [48], have significantly advanced 3D reconstruction pipelines by producing accurate camera poses from multi-view images and videos. More recent models [7, 53] like HLOC [47, 46] and VGGT [56, 57] build on this progress using learning-based features and transformers to handle large-scale matching and pose estimation. While these methods achieve strong performance in estimating camera geometry, they fall short in synchronizing dynamic scenes, as they rely predominantly on static visual cues. In contrast, our approach explicitly decomposes the scene into static and dynamic components, leveraging static cues for pose estimation and dynamic cues from moving objects to perform robust temporal synchronization across views.

**Video Synchronization.** Video synchronization has been explored from multiple perspectives [33, 49, 59, 60, 40, 65]. Geometry-based methods [3, 55, 16, 41, 34, 40], such as those by Albl *et al*. [3] and Li *et al*. [31], estimate temporal offsets using epipolar geometry, but typically assume static scenes or fixed viewpoint. Human-centric approaches use human pose as a synchronization signal, benefiting from its strong visual priors [10, 63, 36, 28, 21], yet these approaches are limited by the accuracy of human pose estimation, the number of people present in the scene, and they struggle in diverse scenarios without prominent human activity. Audio-based approaches [49, 20] use audio cues for synchronization, which can only work in quiet environments and not generally applicable to in-the-wild settings where audio is noisy or unavailable. Learning-based methods like Sync-NeRF [26] jointly optimize camera poses and temporal offsets, but are often constrained to specific environments or object types. Our work overcomes these limitations by leveraging pretrained visual foundation models and framing synchronization as an epipolar-based optimization problem. By reasoning jointly over static structures and dynamic foreground motion, we deliver a generalizable solution for aligning asynchronous, unposed videos in complex real-world scenarios.

## 3 Approach

### 3.1 Problem Formulation

Given a set of $N$ asynchronous videos $\{\mathbf{V}^i\}_{i=1}^N$ capturing the same dynamic scene from different viewpoints, our goal is to synchronize them and recover a globally aligned timestamp. Formally, we aim to estimate a time offset $s^i \in \mathbb{R}$ for each video $i$, to be applied to its original out-of-sync clock time. After synchronization, frames sharing the same clock time will correspond to the exact same moment across all videos.

**Key Insight.** Our key insight lies in the epipolar geometry between two cameras that capture the same scene. In Fig. 2, two cameras with known poses capture the same dynamic scene (e.g., a moving person and their dog). We track and associate a keypoint across both videos (here, the human's hand), yielding a pair of tracklets (yellow and purple). If the videos are synchronized, then for any pair of frames with the same timestamp, the keypoint observations will satisfy epipolar geometry—for example, one keypoint will lie on the epipolar line of its counterpart. Conversely, if the videos are not synchronized, this property does not hold, and the keypoint may deviate from the epipolar line.

Formally, let $\mathbf{x}^i(t)$ and $\mathbf{x}^j(t)$ be a pair of matched 2D tracklets in homogeneous coordinates between cameras $i$ and $j$, forming continuous-time point trajectories and describing the same dynamic 3D point the world. Let $\mathbf{K}_i, \mathbf{K}_j$ denote the known (or estimated) intrinsics, and $\mathbf{T}^i(t), \mathbf{T}^j(t)$ the corresponding

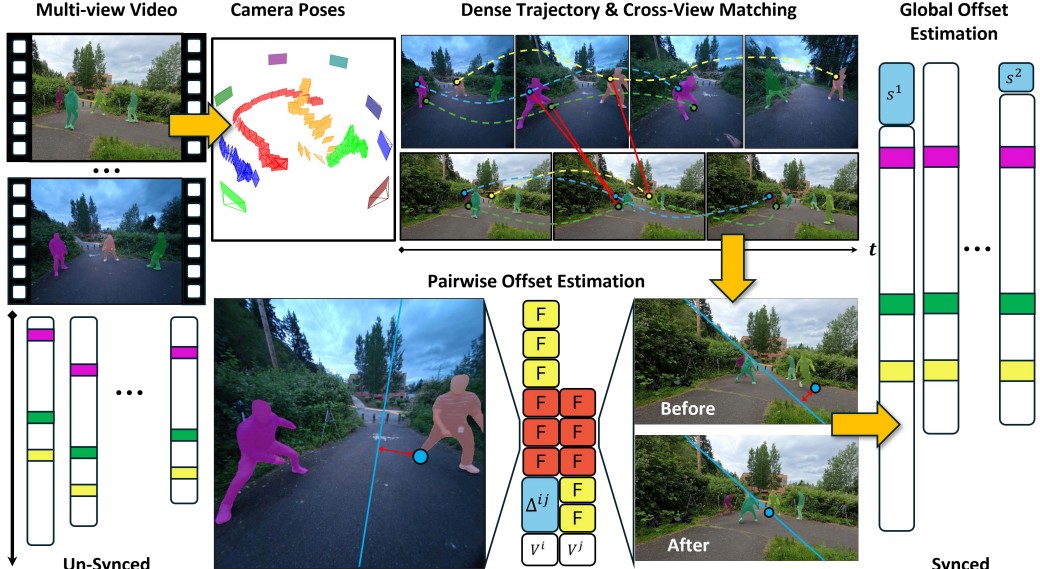

**Figure 3: Proposed framework:** Given unsynchronized videos, VisualSync follows a three-stage pipeline. Stage 0 estimates camera parameters with VGGT [56], dense correspondences with CoTracker3 [23], cross-view matches with MAST3R [30], and dynamic objects with DEVA [9]. In Stage 1, we estimate pairwise frame offsets by minimizing epipolar violations over matched trajectories. Stage 2 globally optimizes individual offsets to produce synchronized videos.

extrinsic trajectories. If the true synchronization offset between cameras $i$ and $j$ is $\Delta$, then the epipolar constraint holds:

$$\left(\mathbf{x}^i(t+\Delta)\right)^\top \mathbf{F}^{ij}_{t+\Delta,t} \mathbf{x}^j(t) \equiv 0; \tag{1}$$

where $\mathbf{F}^{ij}_{t+\Delta,t}$ is the fundamental matrix between camera $i$ at time $t+\Delta$ and camera $j$ at time $t$. Otherwise, it may not be equal to zero.

By leveraging this cue, we formulate an optimization problem that finds the time offset minimizing the epipolar distance of all associated keypoint trajectories for each camera pair with known poses. We will then extend this approach to multi-camera and moving-camera scenarios.

**Problem Formulation.** Inspired by our discussion above, we formulate global synchronization as an energy minimization problem over $\{s^i\}$. Specifically, we aim to find offsets that best align all video pairs by minimizing their pairwise synchronization error:

$$\{s^i\} = \arg\min_{\{s^i\}} \sum_{i<j} E_{ij}(\Delta^{ij}), \quad \text{where } \Delta^{ij} = s^j - s^i \tag{2}$$

Here, $\Delta^{ij}$ denotes the relative temporal offset between videos $i$ and $j$, and $E_{i,j}(\Delta^{ij})$ measures the misalignment error under this candidate offset in terms of the Sampson geometric error between associated tracklet pairs that are covisible between camera $i$ and $j$.

**Pairwise Term.** The key idea of $E_{ij}(\Delta)$ is to quantify how much the paired tracklets violate epipolar geometry. Among various epipolar-error measures, we adopt the Sampson error [19, 35, 45, 44], which approximates the squared Euclidean distance from a point to its corresponding epipolar line. By linearizing the epipolar constraint, it admits a closed-form expression and is computationally efficient for real-world optimization. Detailed derivation are presented in Appendix A. Formally, we write:

$$E_{ij}(\Delta) = \sum_{(\mathbf{x}^i,\mathbf{x}^j)} \sum_{t} \frac{\left(\mathbf{x}^i(t+\Delta)^\top \mathbf{F}^{ij}_{t+\Delta,t} \mathbf{x}^j(t)\right)^2}{\|\mathbf{F}^{ij}_{t+\Delta,t} \mathbf{x}^j(t)\|^2_{1,2} + \|\mathbf{F}^{ij\top}_{t+\Delta,t} \mathbf{x}^i(t+\Delta)\|^2_{1,2}}, \tag{3}$$

where $\mathbf{F}^{ij}_{t+\Delta,t}$ is the fundamental matrix between camera $i$ at time $t+\Delta$ and camera $j$ at time $t$, and $(\mathbf{x}^i(t), \mathbf{x}^j(t))$ are matched continuous-time point-trajectory tracklets between cameras $i$ and $j$. Intuitively, the numerator is the squared algebraic epipolar residual, and the denominator sums the

squared lengths of the two epipolar-line normals. This normalization converts the raw residual into an approximation of the squared point-to-line distance, closely matching the true reprojection error, while remaining a fast and closed-form computation.

## 3.2 Inference

**Challenges.** Minimizing the energy in Eq. (2) poses three challenges. First, the optimization problem is highly non-convex. Second, the formulation is continuous-time, yet observations arrive at discrete frame times. Third, in real-world scenarios, it is difficult to associate dense trajectories across cameras with significantly different viewpoints and to estimate accurate poses for moving cameras.

**Overview.** We address the challenge via a three-stage optimization strategy. **Stage 0** leverages large, pretrained vision models for dense pose-trajectory tracking, feedforward camera-pose and intrinsic estimation, and extreme-viewpoint matching, making energy evaluation tractable. Then we adopt a divide-and-conquer approach to minimize the proposed energy optimization. **Stage 1** performs per-pair, discrete-time surrogate optimizations via exhaustive search to find each camera pair's optimal alignment. **Stage 2** aggregates these pairwise alignments to recover the global, continuous-time offset via solving a robust least square problem.

**Stage 0: Visual Cue Extraction.** Computing $E_{ij}$ defined in Eq. (4) requires camera parameters (intrinsics and poses) and dynamic point trajectory pairs across views. To obtain these, we use VGGT [56] to jointly reasons about all cameras' intrinsics and pose trajectories from static background regions in a common coordinate system. We apply GPT4o, SAM and DEVA [9, 27, 43, 32] and CoTrackerV3 [23] to segment dynamic objects and track dense 2D point trajectories within each video, and we employ MASt3R [30] to match these per-view tracklets across cameras by comparing sampled keyframes, yielding the cross-view correspondences needed for Sampson error evaluation. We provide more details in the appendix.

**Stage 1: Estimating Pairwise Offsets.** To handle the joint-optimization challenge with non-convexity and only discrete visual evidence at each frame, we drop the constraint in Eq. (2) and instead search over a discrete set of offsets $\Delta^{ij} \in \mathcal{S}$, for each camera pair $(i, j)$. In this way, we can independently minimize each pairwise energy term:

$$\forall (i, j): \quad \Delta^{ij*} = \arg \min_{\Delta \in \mathcal{S}} E_{ij}(\Delta), \tag{4}$$

where $\mathcal{S}$ is a finite set of time offsets, step size is determined by the frame rate and range is a hyperparameter.

Note that not all camera pairs $(i, j)$ yield reliable time-offset estimates. In practice, some pairs have minimal temporal overlap, others lack sufficient viewpoint overlap, or our visual cues may be noisy. Using the per-pair estimates in Eq. (4), We discard any pair whose ratio of the optimal energy to the next-best local minimum falls below $0.1$ or more than two local minima found, resulting in $\mathcal{E}$ of reliable pairs for global synchronization.

**Stage 2: Global Offset Esimation.** The goal in this stage is to recover the global offsets $\{s^i\}$ for all videos from the pairwise estimation $\Delta^{ij}$. Since the discrete, imperfect estimates $\Delta^{ij}$ may not admit a solution $\{s^i\}$ perfectly satisfying $s^j - s^i = \Delta^{ij}$ for every pair $(i, j) \in \mathcal{E}$, we formulate a robust least-squares problem:

$$\{s^i\}^* = \arg \min_{\{s^i\}} \sum_{(i,j) \in \mathcal{E}} \rho_\delta \big( s^j - s^i - \Delta^{ij} \big), \tag{5}$$

where $\rho_\delta$ is the Huber loss. We solve this with an iteratively reweighted least squares (IRLS) procedure [11], yielding the final global synchronization offsets $\{s^i\}^*$.

## 4 Experiments
### 4.1 Experimental Setup

**Implementation details.** Given the input videos, we first extract dynamic object categories using GPT [1] and apply Grounded-SAM [43, 30] to obtain initial per-frame segmentations. We then run DEVA [9] to track these instance masks across time, producing temporally consistent segmentations for each moving object. For each tracked instance, we apply CoTracker3 [23] to perform per-instance temporal tracking. To establish cross-view correspondences, we sample keyframes every 10 frames

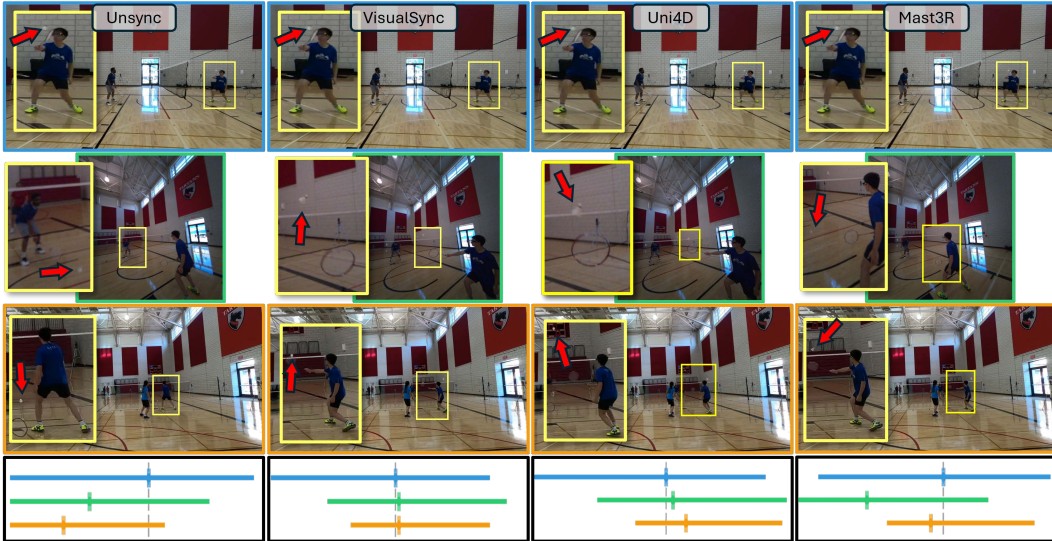

Figure 4: **Qualitative Comparison of synchronization on Egohumans [25] across baselines** We visually assess temporal synchronization by presenting magnified views of the shuttlecock's position across time. In this complex scenario—marked by large temporal discrepancies, a small dynamic element, and moving cameras—Visual Sync achieves the most accurate alignment.

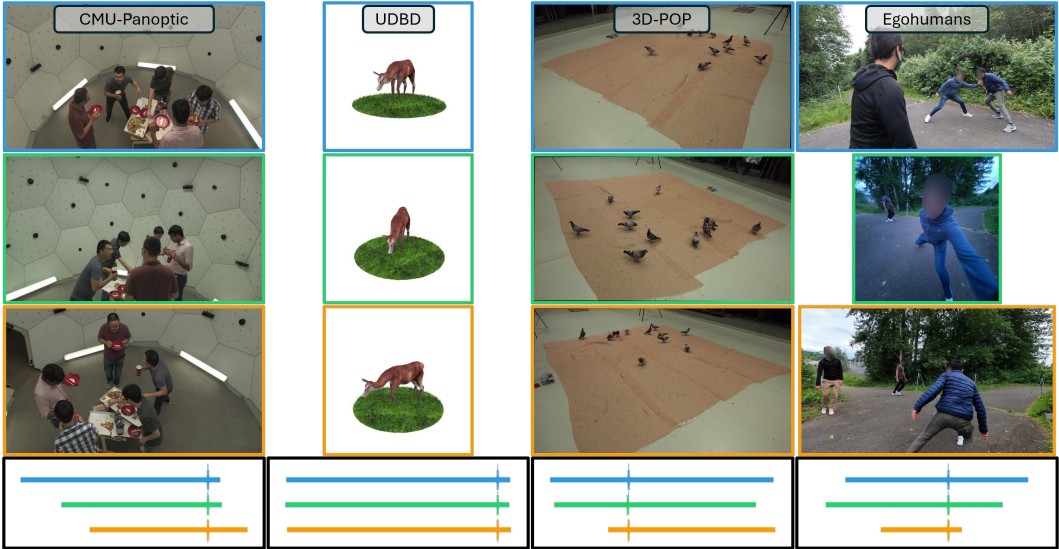

Figure 5: **Qualitative Comparison of Video Sync across datasets.** We show the synchronized videos on CMU-Panoptic, UDBD, 3D-POP and Egohumans dataset. Top 3 rows shows the estimated synchronized time stamps from 3 different views. The bottom row shows synchronized timelines across multiple videos. Our method performs robustly across diverse scenes.

and query Mast3R [30] within the dynamic instance masks, linking tracklets across views. Camera poses are estimated by VGGT [56]. We compute pairwise synchronization energy over the maximum overlapping time window between video pairs and evaluate energy across different offsets. Finally, we select reliable pairs for global synchronization. More details can be found in Appendix A.

**Datasets.** We evaluate our method on a comprehensive suite of multi-view video datasets capturing dynamic scenes. These datasets vary across several dimensions—including camera motion (static vs. dynamic), environment (indoor vs. outdoor), realism (real vs. synthetic), and motion type (human and non-human). CMU Panoptic [22] features a real-world indoor dataset with 30 static cameras captured at 30fps capturing human interactions. Egohumans [25] is a challenging multi-view egocentric and static cameras capturing various sports taking place both indoors and outdoors at different resolutions. 3DPOP [37] is a large scale 2D to 3D posture, identity and trajectory dataset featuring moving

**Table 1: Video Evaluation Results**. For each dataset, we show mean and median errors (ms) for video metrics. We **bold** and underline the best and second best results respectively. Methods with * indicates using GT camera pose as input. Without relying on any GT input, our method achieves the best overall performance across all four datasets, spanning diverse subjects and scenes.

| Method | Egohumans $\delta_{mean}\downarrow$ | $\delta_{med}\downarrow$ | CMU Panoptic $\delta_{mean}\downarrow$ | $\delta_{med}\downarrow$ | 3D-POP $\delta_{mean}\downarrow$ | $\delta_{med}\downarrow$ | UDBD $\delta_{mean}\downarrow$ | $\delta_{med}\downarrow$ |
|---|---|---|---|---|---|---|---|---|
| Uni4D*[62] | 447.4 | 222.1 | 777.9 | 99.9 | 1600.1 | 1265.4 | 103.2 | 25.1 |
| Mast3R[30] | 742.3 | 263.8 | 113.4 | 58.1 | 150.3 | **72.2** | 10.1 | 7.4 |
| Sync-NeRF*[26] | - | - | 919.5 | 866.7 | 1138.9 | 1100.0 | **0.4** | **0.2** |
| **Ours** | **122.1** | **46.6** | **112.6** | **41.5** | **114.7** | 77.8 | 20.2 | 5.9 |

**Table 2: Stage-1 Pairwise Evaluation Results**. For each dataset, we show mean and median errors (ms) for pairwise metrics. We **bold** and underline the best and second best results respectively. Methods with * indicates using GT camera pose as input. Our method outperforms other baselines on both metrics across datasets.

| Method | Egohumans A@100↑ | A@500↑ | CMU Panoptic A@100↑ | A@500↑ | 3D-POP A@100↑ | A@500↑ | UDBD A@100↑ | A@500↑ |
|---|---|---|---|---|---|---|---|---|
| Uni4D*[62] | 23.8 | 49.4 | **32.3** | **60.7** | 0.9 | 9.5 | 46.2 | 74.1 |
| Mast3R[30] | 24.3 | 50.4 | 29.6 | 49.8 | 15.7 | 69.1 | 77.8 | 95.4 |
| Sync-NeRF*[26] | - | - | 3.0 | 13.8 | 0.0 | 8.2 | **86.7** | **97.35** |
| **Ours** | **33.9** | **55.8** | 26.0 | 51.2 | **33.3** | **69.3** | 82.1 | 94.3 |

pigeons.Unsynchronized Dynamic Blender Dataset (UDBD) is a synthetic toy example created with dynamic blender assets used in SyncNeRF [26].

To prepare these multi-view datasets for multi-video synchronization, we take subsequences of each video while ensuring that they all have a common overlap. Each sequence is roughly 10 seconds long, with a random cropping of around 2-3 seconds from the front and back to simulate offsets and unsynchronized videos. These offsets are used for evaluative purposes.

**Baselines.** For baselines, we explore leading methods using different strategies for multi-video synchronization. Following Uni4D [62], we adopt a geometric approach using metric depth estimation to compute Chamfer distances between projected dynamic pixels. Ground-truth camera poses are used to triangulate scene points and resolve per-image scale ambiguity. For a learning-based approach, we use Mast3r [30], which leverages attention and confidence maps shown to capture motion and rigidity [8]. We compute energy for each offset as the mean confidence between keyframe pairs within dynamic masks. Sync-NeRF [26] incorporates temporal offsets into photometric optimization. We adapted its codebase for varying intrinsics, but excluded Egohuman due to egocentric camera challenges. For Uni4D and Mast3r, we compute pairwise offsets followed by global optimization, similar to our method.

**Metrics.** We evaluate both pairwise and per-video offset performance across all datasets. For pairwise evaluation, predicted offsets are compared with ground-truth relative offsets between every pair of cameras within the same dynamic scene. We report the AUC for error thresholds at 100ms (A@100) and 500ms (A@500) respectively. For video evaluation, we compute a single offset per-video while fixing the offset for the same reference camera. We report both the mean ($\delta_{mean}$) and median ($\delta_{med}$) error. Since our datasets have different FPS, we report our results in milliseconds. The mean synchronization error ( 100 ms) is heavily influenced by a small number of extremely challenging camera views, as we did not exclude any views from our evaluation. Typically, the mean error is more sensitive to a few challenging camera views, as we did not exclude any from evaluation.

## 4.2 Results

**Quantitative.** Our analysis of offset results across diverse datasets and baselines is detailed in Tab. 1 (video synchronization) and Tab. 2 (pairwise synchronization). Notably, on the challenging EgoHumans dataset [25], VisualSync demonstrates superior performance, achieving successful video synchronization with a median error of just 46.6 milliseconds. The geometric approach, Uni4D, exhibits strong performance on datasets where dynamic objects are in closer proximity to the camera.

**Table 3: Ablation of key components on Egohumans dataset**. We **bold** and underline the best and second best results respectively. The *1st* two rows show the oracle performance leveraging GT information as input. The *2nd* block compares different camera pose estimations, the *3rd* block compares different energy terms, the *4th* block compare different solvers for global optimization. Our proposed pipeline achieves the best overall performance.

| Design Choices | | | | Solver | Pairwise | | Video | |
|---|---|---|---|---|---|---|---|---|
| Segmentation | Correspondence | Camera | Energy | | A@100↑ | A@500↑ | $\delta_{mean}\downarrow$ | $\delta_{med}\downarrow$ |
| GT | GT | GT | Sampson | IRLS | 94.8 | 97.9 | 11.3 | 2.0 |
| DEVA | CoTracker+Mast3R | GT | Sampson | IRLS | 56.0 | 85.0 | 72.4 | 28.6 |
| DEVA | CoTracker+Mast3R | hloc | Sampson | IRLS | 38.2 | 68.8 | 199.5 | 75.3 |
| DEVA | CoTracker+Mast3R | ransac | Inlier | IRLS | 20.1 | 39.5 | 1656.5 | 1544.8 |
| DEVA | CoTracker+Mast3R | vggt | Cosine | IRLS | 28.0 | 61.1 | 239.8 | 94.6 |
| DEVA | CoTracker+Mast3R | vggt | Algebraic | IRLS | 32.6 | 63.6 | 167.3 | 57.9 |
| DEVA | CoTracker+Mast3R | vggt | Epipolar | IRLS | 36.2 | 69.4 | 125.0 | **35.4** |
| DEVA | CoTracker+Mast3R | vggt | Sampson | LS | 20.3 | 56.7 | 205.9 | 118.0 |
| DEVA | CoTracker+Mast3R | vggt | Sampson | IRLS | **40.2** | **73.4** | **122.1** | 46.6 |

This suggests that more accurate metric estimations in these scenarios directly translate to improved alignment results. Conversely, Uni4D's performance significantly deteriorates on the 3D-POP dataset [37], where small, distant dynamic objects (pigeons) likely introduce inconsistencies in multi-view metric estimations. The energy landscape of Uni4D is also noisy and there is no clear cues to remove spurious pairwise results, resulting in worse global synchronization. The optimization-based method, Sync-NeRF, jointly optimizes temporal offsets and photometric loss. However, consistent with findings in related work [10], Sync-NeRF struggles to calibrate more complex dynamic scenes beyond UDBD [26] in settings of large offsets like ours, likely due to a lack of strong priors and effective offset initialization. Interestingly, the learning-based approach, Mast3R [30], showcases surprisingly strong generalization capabilities. Despite not being explicitly trained for this task, it outperforms the other two baselines on several datasets in both pairwise and video synchronization evaluations. However, its performance on the EgoHumans dataset [25] is notably weaker, potentially indicating a limitation in handling challenging egocentric views and motion blur.

**Qualitative.** We present qualitative comparisons on the EgoHumans dataset [25] against Uni4D and Mast3R in Fig. 4. As shown in Fig. 4, both methods struggle with highly dynamic motions and egocentric–third-person alignment. The timeline below shows ground-truth key events and sequence lengths. We further demonstrate our method on four datasets (Fig. 5) and in-the-wild sports footage (NBA, EFL) (Fig. 6), effectively handling rapid motion, zooms, and dynamic camera movements across diverse scenes. Additional qualitative results can be seen in Fig. 13 and Fig. 14.

### 4.3 Analysis

**Key components.** We evaluate the contribution of key components in our framework through an ablation study on the EgoHumans dataset, as shown in Tab. 3. Note A@100 and A@500 are intermediate metrics across all video pairs, including those with opposite viewpoints or no temporal overlap. The **first block** reports oracle results using ground-truth segmentation, camera poses, and correspondences for all video pairs—regardless of overlap—demonstrating near-perfect performance with perfect inputs and serving as an upper bound. The **second block** examines different camera pose estimation methods. Our framework performs consistently across alternatives, with VGGT achieving the best results. Compared to the 28.6ms oracle result (2nd row), using estimated camera poses from VGGT achieves a 46.6ms median error, demonstrating the robustness of our approach under imperfect pose inputs and its effectiveness in practical conditions. We further report camera pose and synchronization results (relative angular error in rotation and translation following VGGT) across randomly selected EgoHumans sports videos in Tab. 7. The **third block** ablates various pairwise energy terms used for synchronization. We include a baseline method inspired by [3], which relies solely on dynamic tracklets and uses RANSAC to compute inlier matches as the pairwise energy metric. In contrast, our method leverages both static background and camera pose information. We also evaluate three geometric energy terms—*cosine error*, *algebraic error* [29], and *symmetric epipolar distance*—which are detailed in Appendix A. Among all energy terms, the *Sampson error* performs best, as it explicitly models noise in the tracklets and provides a lower bound

**Table 4: Ablation of input settings, spurious pair detection, and stage contribution on the Egohumans dataset.** The first *3* rows ablate number of input pairs. RST denotes a Random Spanning Tree using only the minimal number of pairs needed to form connectivity. For each setting, we run 10 times and report the mean and variance for each metric. Pairwise metrics in $*$ are computed after global sychronization except for *4th* row. In *5th* row, we show the importance of removing spurious pairs. Our method achieves comparable performance even only using $50\%$ of pairs input.

| Input Setting | | Stage | Pairwise* | | Video | |
|---|---|---|---|---|---|---|
| Pairs Ratio | Spurious Det. | | A@100↑ | A@500↑ | $\delta_{mean}$↓ | $\delta_{med}$↓ |
| RST | ✓ | Full | $19.5 \pm 1.9$ | $43.2 \pm 3.1$ | $436.4 \pm 63.1$ | $130.0 \pm 24.5$ |
| 50% | ✓ | Full | $28.9 \pm 0.5$ | $59.8 \pm 1.0$ | $212.6 \pm 9.5$ | $70.7 \pm 1.3$ |
| 80% | ✓ | Full | $35.4 \pm 0$ | $69.2 \pm 0$ | $144.4 \pm 0$ | $\mathbf{44.7 \pm 0}$ |
| 100% | ✓ | Stage-1 | 33.9 | 55.8 | - | - |
| 100% | ✗ | Full | 30.7 | 58.1 | 371.4 | 111.5 |
| 100% | ✓ | Full | **40.1** | **73.4** | **122.1** | 46.6 |

**Table 5: Ablation of varying frame rates** on the CMU Panoptic dataset. We keep 30 fps for the constant setting and downsample them to 5–30 fps for the varying setting, achieving similar performance without any pipeline changes.

**Table 6: Ablation of low frame Rates** on the CMU Panoptic dataset. Downsampling from 30 fps to 15 fps causes a slight performance drop due to reduced temporal overlap, yet the method remains robust under the low-fps setting.

| Input FPS | $\delta_{mean}$↓ | $\delta_{med}$↓ |
|---|---|---|
| Constant | 112.6 | 41.5 |
| Varying | 103.9 | 51.5 |

| Input FPS | $\delta_{mean}$↓ | $\delta_{med}$↓ |
|---|---|---|
| 30 | 112.6 | 41.5 |
| 15 | 157.2 | 45.6 |

on the true epipolar error through a linear approximation under noisy estimates. The **last block** compares different solvers. A least-squares solver performs reasonably but is sensitive to outliers. Our IRLS-based solver achieves better accuracy by down-weighting unreliable estimates. Overall, our configuration achieves the best performance, validating the effectiveness of each design choice.

**Input and different stage contributions.** Tab. 4 evaluates the impact of input pair selection, spurious pair removal, and multi-stage optimization. In the *first block*, we ablate the number of input pairs. RST uses only the minimal set to form a connected graph, yet achieves $<150$ ms median error. Even with 50% of pairs, performance remains close to the full setting, showing robustness to limited input. The *second block* highlights the benefits of spurious pair filtering and two-stage optimization. By exploiting energy landscape structure, our method filters unreliable pairs and improves global accuracy. The final row, using the full pipeline, yields the best results.

**Input frame rate.** To evaluate the robustness of our method under different temporal conditions, we conduct two ablation study on the CMU Panoptic dataset [22]. In Tab. 5, we test synchronization across videos with varying frame rates (2nd row), sampling each video between 5 fps and 30 fps. Our method is applied directly without any pipeline changes, achieving 51.5 ms performance—comparable to 41.5 ms at the original 30 fps setting—demonstrating strong adaptability to frame rate variations in real-world data. In Tab. 6, we test robustness to low frame rates by downsampling videos from 30 fps to 15 fps. While performance slightly degrades due to reduced temporal overlap, the method remains accurate, demonstrating resilience under sparse sampling. Notably, our preprocessing modules (Co-Tracker [23] and DEVA [9]) are optimized for high frame rates, making 15 fps particularly challenging.

**Runtime.** Motion segmentation and VGGT [56] pose estimation run at 0.3 s and 0.35 s per frame, respectively. Tracking [23] takes 120 s per 10 s video, and Mast3R [30] takes 60 s per video pair. Energy evaluation and global sync are efficient, taking under 10 s and 1 s per pair, respectively. Although our method is $\mathcal{O}(N^2)$ in the number of videos, we show in Tab. 4 that using only 50% of pairs or a Random Spanning Tree (RST) offers a good trade-off. All runtimes are measured on a single A6000 GPU. On the CMU Panoptic dataset (15 videos, 200 frames each), our method takes about 3.3 hours—comparable to Uni4D (3.9 hrs) and Sync-NeRF (4.2 hrs), though slower than MAST3R (1.2 hrs). Efficiency can be further improved with lightweight modules and additional computing resources with parallel preprocessing, making the method practical for offline multi-camera applications such as sports analysis, film production, and surveillance.

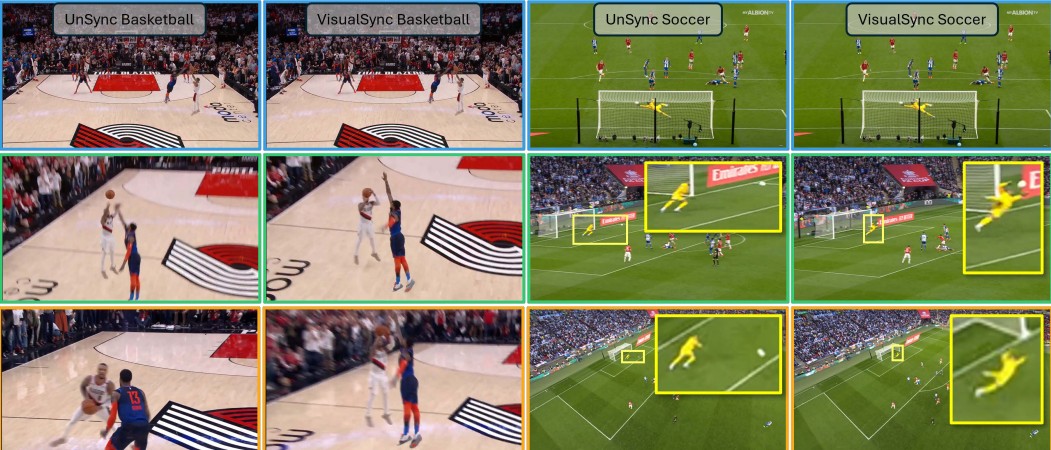

**Figure 6: Qualitative Synchronization of VisualSync on In-the-Wild Sports Videos.** We showcase VisualSync on challenging multi-view sports footage with large camera motions, motion blur, and zoom variations. Despite these real-world challenges, our method achieves accurate synchronization. In the absence of ground-truth alignments, we qualitatively verify accuracy through the precise alignment of key events (e.g., ball release, contact) across views.

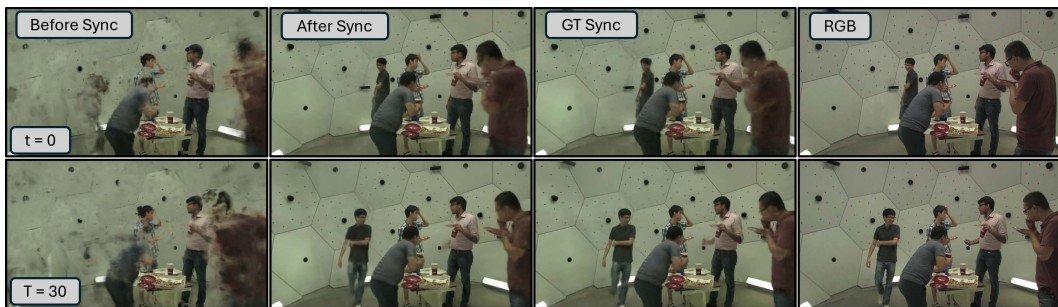

**Figure 7: K-Plane Rendering on VisualSync Results.** We train K-Planes on CMU Panoptic [22] multi-view videos for novel view synthesis. Unsynchronized inputs (1st) produce blurry results, while our synchronized outputs (2nd) are sharp and comparable to ground-truth sync (3rd). The 4th column shows real images. Our method enables high-quality synthesis from unsynchronized inputs.

**Limitations.** Our method has three main limitations. First, it requires a subset of reliable camera poses (not necessary for the entire sequence). Second, it can't handle clips containing non-uniform motion speeds—for example, videos that alternate between slow-motion and fast-motion segments. Third, the pairwise estimation step scales quadratically ($\mathcal{O}(N^2)$) with the number of videos, which can affect efficiency in large-scale setups. Failure case analysis of each module is shown in Fig. 12.

### 4.4 Application

Video synchronization unlocks downstream applications like multi-view dynamic scene reconstruction. As demonstrated in Fig. 7, directly applying K-Planes [17] to unsynchronized data yields unsatisfactory novel view renderings. In contrast, our Video Sync approach enables significantly improved novel view synthesis, achieving results comparable to those obtained with ground truth synchronized video. This demonstrates that our method can serve as a fundamental tool for downstream applications such as novel-view synthesis in real, multi-view unsynchronized dynamic world.

## 5 Conclusion

We presented *VisualSync*, a robust framework for synchronizing unposed, unsynchronized multi-camera videos with millisecond accuracy. By minimizing epipolar error over dense correspondences, it recovers precise time offsets across diverse scenes and motions. Experiments on four datasets show VisualSync outperforms recent methods. Our approach contributes a practical step toward enabling dynamic multi-view video motion understanding and related downstream tasks.

## Acknowledgements

This project is supported by NSF Awards #2525287, #2404385, #2414227, #2340254, #2312102, and #2331878, the IBM IIDAI Grant, and an Intel Research Gift. We greatly appreciate the NCSA for providing computing resources.

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

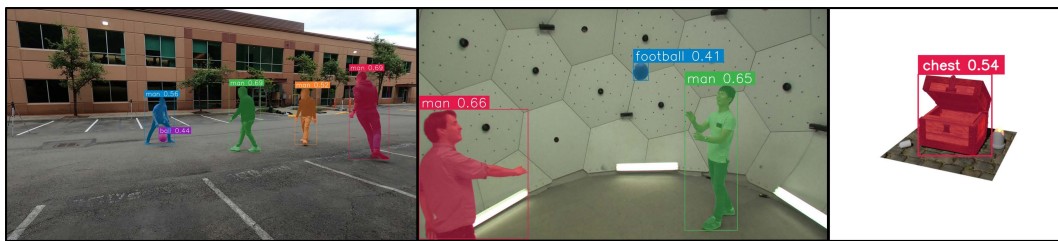

**Figure 8: Grounding-DINO [32] + SAM2 [42] Segmentation results** We visualize Grounding-DINO's proposed bounding box given the dynamic labels produced by GPT4o, along with SAM2 segmentation results with confidence scores. Note that objects that are generally not dynamic (eg. basketball, football, chest) is identified as dynamic in these specific scenes due to inputting video frames into GPT4o.

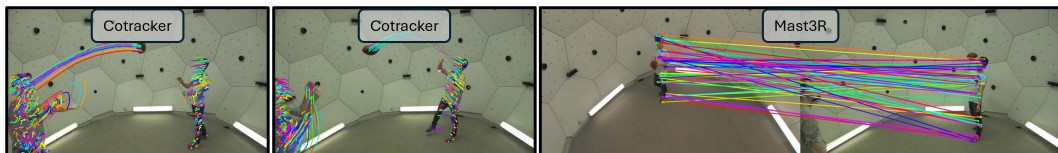

**Figure 9: Visualization of Cotracker and Mast3R correspondences** We visualize actual spatial-temporal correspondences predicted using CotrackerV3 (temporal) and Mast3R (spatial).

## A Implementation Details

**Motion Segmentation.** For dynamic object segmentation, we follow the pipeline of Uni4D [62] with key modifications. Unlike Uni4D, which relies on Recognize Anything [64] and LLM filtering, we directly feed video frames into GPT-4o to identify dynamic classes. The GPT4o Prompt used for automatic motion segmentation is shown in Fig. 11. Every 20th frame is sampled as input, and the detected classes guide GroundingDINO [32] to generate bounding boxes, which then prompt SAM 2 for precise segmentation masks. As shown in Fig. 8, this pipeline achieves robust dynamic object segmentation, with DEVA [9] applied afterward for temporally consistent motion segmentation.

**Spatial-Temporal Correspondence.** To capture motion trajectories of dynamic objects, we apply CoTracker [24] to each video $i$ within its dynamic region $\{\mathbf{M}_t^i\}$, producing a set of 2D tracklets $\{\mathbf{X}^i\}$. Each tracklet $\mathbf{x}^i = \{\mathbf{x}^i(t)\}$ represents the observed image-space trajectory of a dynamic 3D point over time. To establish spatio-temporal correspondences across views, we perform cross-view matching using Mast3R [30]. For each tracklet $\mathbf{x}^i$ in video $i$, we seek its matching tracklet $\mathbf{x}^j$ in another video $j$. To ensure robust matching under asynchronous capture, we sample a subset of keyframes from each tracklet and compute pairwise similarity between all keyframe pairs across views. For each pair of sampled frames, we construct a candidate tracklet pair $(\mathbf{x}^i, \mathbf{x}^j)$ from the correspondences obtained by Mast3R [30] in the two views. The visualization output is shown in Fig. 9.

**Correspondence Filtering.** To further suppress noise, we leverage instance segmentation matching from DEVA to filter correspondences across video pairs. We first construct a pairwise matching matrix by counting correspondences between each instance pair over a sampled subset of frames. Bipartite matching is then applied to obtain optimal one-to-one assignments between instances in the two videos. To ensure reliability, we discard matched instance pairs with fewer than 100 correspondences. After this instance-level matching, we retain only Mast3R correspondences whose endpoints belong to the same matched instance. All remaining correspondences are aggregated into a unified set of spatio-temporal matches, which serve as input to our energy-based synchronization process.

**Camera Parameters.** We use VGGT [56] to extract camera poses and intrinsics in our preprocessing pipeline. For static cameras, we feed only the first frame as input, while for dynamic cameras, all available frames are used to estimate poses and intrinsics. To manage memory constraints, we subsample dynamic sequences to ensure that every offset computation has overlapping frames with predicted camera poses. The outputs of VGGT include per-frame extrinsics $\{\mathbf{P}_t^i \in \mathrm{SE}(3)\}$ and intrinsics $\mathbf{K}_t^i$ for each video $i$, where $\mathbf{P}_t^i = [\mathbf{R}_t^i \mid \mathbf{t}_t^i]$. These parameters are then used to compute relative poses and fundamental matrices. Example output is shown in Fig. 10.

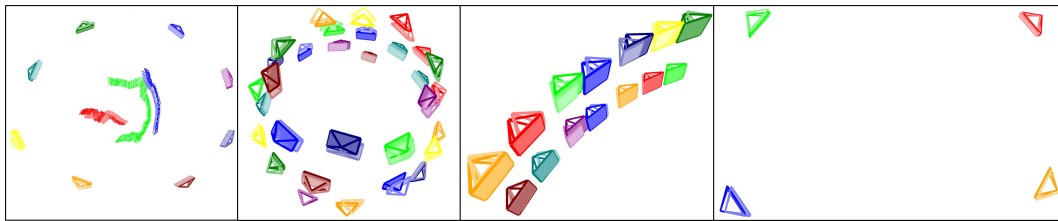

**Figure 10: VGGT [56] Predicted Camera poses compared to GT poses for all datasets** We visualize VGGT predicted camera poses and ground truth camera poses for Egohumans [25], Panoptic [22], UDBD [26], and 3D-POP [37] respectively. Different colors represent different multi-view cameras, while the corresponding lighter palette represents ground truth camera poses.

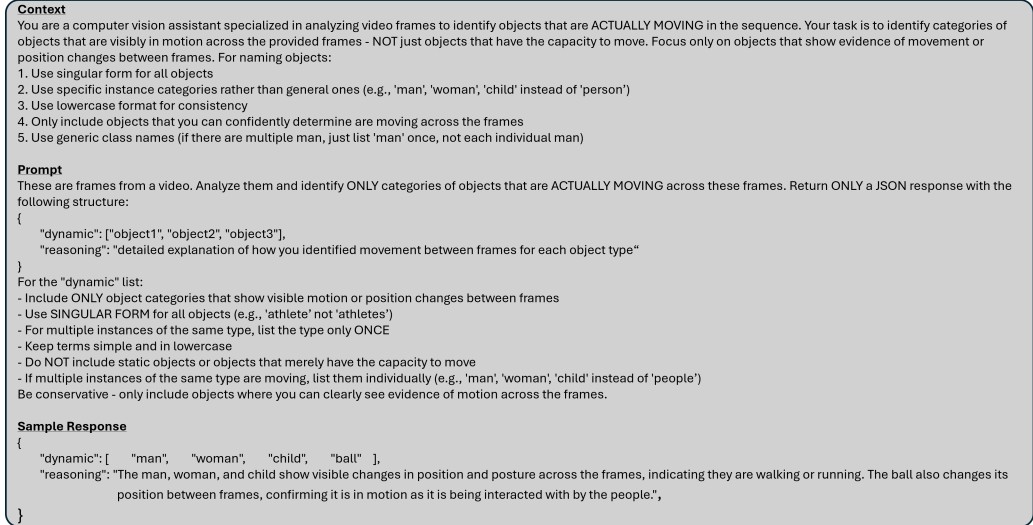

**Figure 11: GPT4o Prompt used for automatic motion segmentation** We sample every 20th frame from our video and input to GPT4o with the following context and prompt to identify motion classes to be given to GroundingDINO module for robust video motion segmentation.

**Sampson Error.** We use the Sampson error in Sec. 3.1 to quantify epipolar constraint violations. Below, we analyze it as a linearized approximation of the true epipolar distance. Let $\mathbf{x}_t = [\mathbf{x}^i(t + \Delta); \mathbf{x}^j(t)]$ denote a noisy spatial-temporal correspondence at time $t + \Delta$ in video $i$ and $t$ in video $j$. Let $\mathbf{z}_t$ be the underlying clean correspondence. To quantify the deviation of the noisy observation $\mathbf{x}_t$ from satisfying the epipolar constraint, we define the energy $\mathcal{E}$ as:

$$\mathcal{E}^2 = \min_{\mathbf{z}_t} \quad \|\mathbf{z}_t - \mathbf{x}_t\|^2 \tag{6}$$

$$\text{s.t.} \quad C(\mathbf{z}_t) = 0 \tag{7}$$

This energy measures the minimal correction required to project $\mathbf{x}_t$ onto the epipolar manifold defined by $C(\mathbf{z}_t) = 0$. The constraint function $C(\mathbf{z}_t)$ evaluates the epipolar consistency between two views:

$$C(\mathbf{z}_t) = \mathbf{x}^i(t + \Delta)^\top \mathbf{F}^{ij}_{t+\Delta,t} \mathbf{x}^j(t) \tag{8}$$

where $\mathbf{F}^{ij}_{t+\Delta,t}$ is the fundamental matrix at frames $t + \Delta$ and $t$.

Since this constrained optimization is nonlinear and difficult to solve directly, we approximate it by linearizing the constraint around $\mathbf{x}_t$. Using a first-order Taylor expansion, we obtain:

$$C(\mathbf{z}_t) \approx C(\mathbf{x}_t) + \mathbf{J}_t(\mathbf{z}_t - \mathbf{x}_t) \tag{9}$$

where $\mathbf{J}_t$ is the Jacobian of $C$ with respect to $\mathbf{x}_t$. Solving for the optimal correction yields the Sampson approximation of $\mathcal{E}^2$, which provides a first-order lower bound on the original energy.

$$\mathcal{E}^2_{Sampson} = \frac{\left(\mathbf{x}^i(t + \Delta)^\top \mathbf{F}^{ij}_{t+\Delta,t} \mathbf{x}^j(t)\right)^2}{\|\mathbf{F}^{ij}_{t+\Delta,t} \mathbf{x}^j(t)\|^2_{1,2} + \|\mathbf{F}^{ij\top}_{t+\Delta,t} \mathbf{x}^i(t + \Delta)\|^2_{1,2}}, \tag{10}$$

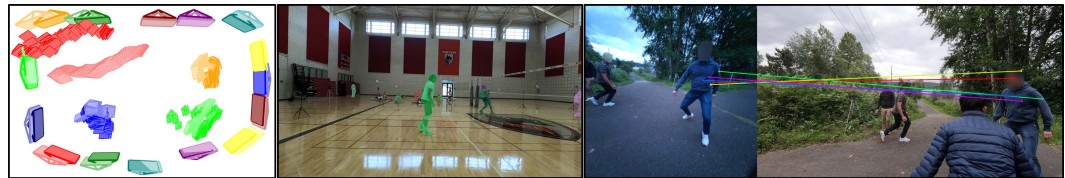

**Figure 12: Failure Case Visualizations** We visualize failure cases across our camera pose, motion segmentation, and spatial correspondence modules. Specifically, observe the incorrect predicted camera poses for the dynamic camera (red) against the ground truth (pink). Background individuals are occasionally mis-segmented for motion segmentation, and some segmentations appear fragmented. Furthermore, Mast3R sometimes generates incorrect correspondences within dynamic masks.

**Table 7: Camera pose error vs. synchronization accuracy.** Even with large rotation and translation errors in camera pose estimates (relative angular errors from VGGT), our method maintains low synchronization errors (*e.g.*, 9.6 ms for *Fencing* and 19.4 ms for *Tagging*). The higher error in *Tennis* stems from distant camera placement with minimal observable motion.

| Metric | Fencing | Volleyball | LegoAssemble | Badminton | Tagging | Basketball | Tennis |
|---|---|---|---|---|---|---|---|
| Cam Rot Err $\downarrow$ | 10.9 | 8.5 | 3.1 | 10.2 | 5.8 | 4.0 | 2.7 |
| Cam Trans Err $\downarrow$ | 14.0 | 14.6 | 7.1 | 11.8 | 9.1 | 12.2 | 13.9 |
| $\delta_{med} \downarrow$ | 9.6 | 30.8 | 38.3 | 34.6 | 19.4 | 27.5 | 113.6 |

**Other Energy Terms.** Our ablation study also compares other three distinct geometric energy terms: symmetric epipolar distance, cosine error, and algebraic error. Following the definitions proposed by Terekohov et al. [54], these terms are formally defined as:

$$\mathcal{E}_{Epipolar}^2 = \frac{|(\mathbf{x}^i(t+\Delta))^\mathsf{T}\mathbf{F}_{t+\Delta,t}^{ij}\mathbf{x}^j(t)|^2}{\|\mathbf{F}_{t+\Delta,t}^{ij}\mathbf{x}^j(t)\|_{1,2}^2} + \frac{|(\mathbf{x}^i(t+\Delta))^\mathsf{T}\mathbf{F}_{t+\Delta,t}^{ij}\mathbf{x}^j(t)|^2}{\|(\mathbf{F}_{t+\Delta,t}^{ij})^\mathsf{T}\mathbf{x}^i(t+\Delta)\|_{1,2}^2} \tag{11}$$

$$\mathcal{E}_{Cosine}^2 = \frac{|(\mathbf{x}^i(t+\Delta))^\mathsf{T}\mathbf{F}_{t+\Delta,t}^{ij}\mathbf{x}^j(t)|^2}{\|\mathbf{x}^i(t+\Delta)\|^2\|\mathbf{F}_{t+\Delta,t}^{ij}\mathbf{x}^j(t)\|^2} + \frac{|(\mathbf{x}^i(t+\Delta))^\mathsf{T}\mathbf{F}_{t+\Delta,t}^{ij}\mathbf{x}^j(t)|^2}{\|(\mathbf{F}_{t+\Delta,t}^{ij})^\mathsf{T}\mathbf{x}^i(t+\Delta)\|^2\|\mathbf{x}^j(t)\|^2} \tag{12}$$

$$\mathcal{E}_{Algebraic} = |(\mathbf{x}^i(t+\Delta))^\mathsf{T}\mathbf{F}_{t+\Delta,t}^{ij}\mathbf{x}^j(t)| \tag{13}$$

As shown in [44, 54] and our experiments in Tab. 3, the Sampson error is more robust to input noise. In our setting, we extend the point-wise Sampson error to a spatial-temporal formulation by considering tracklets as input.

# B    Failure Case

VisualSync depends on several upstream modules and is thus sensitive to their prediction errors. As shown in Fig. 12, inaccuracies in camera pose estimation, motion segmentation, or correspondence matching can propagate through the pipeline and introduce error in synchronization. However, most unreliable cases could be detected by analyzing the pairwise energy landscape. Estimates with ambiguous or low-confidence minima are discarded to prevent degradation of global synchronization. Despite occasional failures, our robust preprocessing effectively limits their impact, enabling Visual-Sync to maintain strong performance for in-the-wild videos. Its modular design also ensures that improvements to individual components directly enhance overall synchronization quality.

# C    Additional Results

**Camera pose error vs. synchronization accuracy.** We report camera pose and synchronization results (relative angular errors in rotation and translation following VGGT) across randomly selected EgoHumans sports videos in Tab. 7, demonstrating the robustness of our approach under varying pose estimation quality. Even with large pose errors, our method maintains low synchronization errors (e.g., 9.6 ms for Fencing and 19.4 ms for Tagging), indicating that pose noise does not directly lead to synchronization failure.

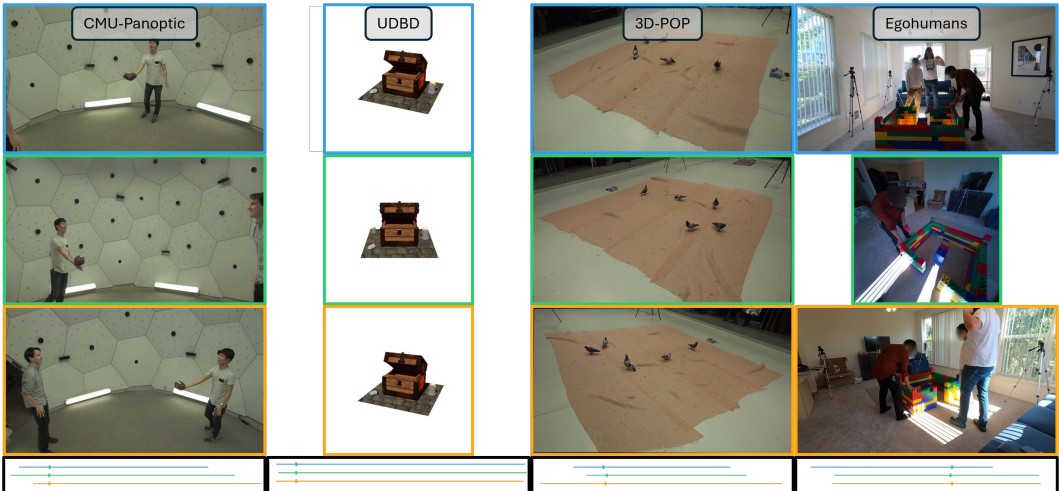

**Figure 13: Qualitative Comparison of VisualSync across datasets** We show the synchronized videos on CMU-Panoptic, UDBD, 3D-POP and Egohumans dataset using VisualSync. The top 3 rows show the estimated synchronized time stamps from 3 different views. The bottom row shows synchronized timelines between multiple videos. Our method performs robustly across diverse scenes.

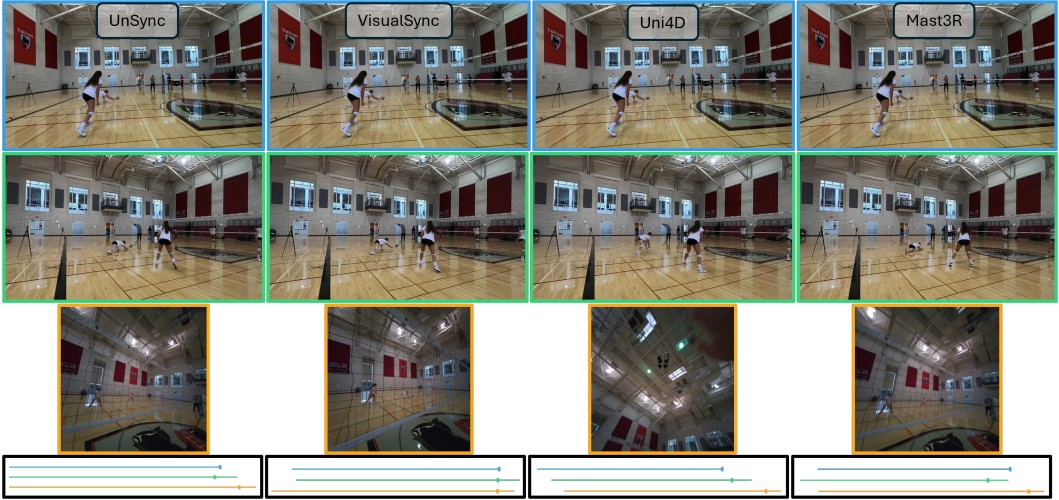

**Figure 14: Qualitative Comparison of synchronization on Egohumans across baselines** We visualize synchronization results in the challenging volleyball sequence in Egohumans. Notice that VisualSync achieves the most accurate alignment even for egocentric views (orange highlight).

**Qualitative results.** To further demonstrate VisualSync's capabilities, we present additional qualitative results across 4 datasets, specifically Egohumans [25], Panoptic [22], UDBD [26], and 3D-POP [37], in Fig. 13. We also provide comprehensive comparisons with baselines Uni4D [62] and Mast3R [30] in Fig. 14. Note that we excluded Sync-NeRF [26] for comparison as it struggled to produce meaningful offset predictions beyond its simpler native UDBD dataset. For visualization, we show three selected camera views. The first view (blue highlight) serves as the reference, and corresponding frames from other views are aligned accordingly. The timeline below depicts ground-truth keyframe alignments, marking the synchronized frame positions across camera sequences. These qualitative results demonstrate VisualSync's strong synchronization performance across datasets and baselines. Full video visualizations can be found in the supplementary material or on the project page at https://stevenlsw.github.io/visualsync.

