# OpenReview forum: "Visual Sync: Multi‑Camera Synchronization via Cross‑View Object Motion"
_NeurIPS.cc/2025/Conference — NeurIPS 2025 poster_

### Official Review · Reviewer_dtE8 · 2025-06-28

**Clarity:** 3
**Significance:** 2
**Originality:** 3
**Rating:** 4
**Confidence:** 3

**Summary:**

This paper presents VisualSync, a training-free framework for synchronizing unsynchronized videos captured from multiple cameras. It formulates synchronization as an epipolar-based optimization problem and leverages pretrained visual foundation models to construct a robust three-stage optimization pipeline. Experimental results demonstrate superior performance across four diverse datasets, consistently outperforming strong baselines including SyncNeRF, Uni4D, and Mast3R.

**Questions:**

Please see the weaknesses section, especially the heavy reliance on the other models and the efficiency.

**Ethical Concerns:**

["NO or VERY MINOR ethics concerns only"]

**Final Justification:**

My final score for this paper is 'Borderline Accept'. The paper presents a neat idea, well elaborated and tested.

**Limitations:**

yes

**Quality:**

2

**Strengths And Weaknesses:**

Paper Strengths
1. Novel Insight.  The idea of leveraging epipolar constraints to achieve video synchronization is new to me.
2. Complete Systematic Design. A training-free, well-designed three-stage inference pipeline, including visual cue extraction, estimating pairwise offsets, and global offset estimation, is proposed.
3. Extensive Experiments. The method is evaluated on four diverse datasets (CMU Panoptic, EgoHumans, UDBD, and 3D-POP) with detailed ablations.


Weaknesses
1. Heavy reliance on external models:
The proposed pipeline depends on several pretrained models, including GPT, Grounded-SAM, DEVA, CoTracker3, Mast3R, and VGGT, to extract visual cues. This makes the system overly reliant on the robustness of these components and also leads to a too complex, inefficient pipeline. The performance is particularly sensitive to VGGT’s pose estimation quality, which is not always reliable, especially under challenging viewpoints or motion blur.

2. Computational cost and scalability:
As noted in Lines 248–250, components like CoTracker3 and Mast3R are already computationally expensive. Combined with the method’s O(N²) pairwise synchronization strategy, this severely limits scalability, especially for latency-sensitive or large-scale applications. Furthermore, the paper lacks a runtime or efficiency comparison with existing baselines or competitors, which is important for assessing practical usability.

3. Lack of failure case analysis:
The paper briefly mentions potential failure cases (e.g., fast camera motion, small dynamic objects), but does not provide qualitative examples or a detailed discussion. Including visual or diagnostic results for such scenarios would help clarify the method’s limitations and robustness boundaries.

---

> ### Author Rebuttal · Authors · 2025-07-31
>
> ### a) Reliance on external models
>
> While our pipeline integrates state-of-the-art tools such as VGGT, MAST3R, and CoTracker, our core contribution lies in the robust energy formulation and inference framework, which is modular and robust to upstream noise.
>
> Our **energy function value itself serves as a robust indicator** for detecting unreliable outputs from preceding models. As shown in **Tab. 4 (5th and bottom row)** , our method leverages the energy landscape to **actively filter out spurious pairs**, allowing the system to tolerate failures in individual components—especially noisy pose predictions from VGGT.. This design ensures that **localized errors** (e.g., due to motion blur or egocentric viewpoints) do **not compromise overall synchronization**.
>
>
> Crucially, our method does **not require accurate poses for every frame**. In practice, a **subset of frames with reliable pose estimates is sufficient for robust synchronization**, thanks to our energy-based optimization and filtering of spurious pairs. As a result, our method gracefully handles imperfect predictions and remains robust in challenging scenarios.
>
> *EgoHumans* dataset presents severe challenges due to **egocentric viewpoints** and **abrupt head motion** (**see Supp videos for visualization of these challenges**). As shown in **Tab. 3 (2nd and bottom row)**, using estimated poses on the challenging *EgoHumans* dataset yields a **median error of 46.6 ms**, close to the **28.6 ms** oracle result with ground-truth poses.
>
> We report **camera pose and synchronization results** (relative angular error in rotation and translation following VGGT)  across randomly selected EgoHumans sports videos below, demonstrating the robustness of our approach under varying pose estimation qualities. Even with large rotation and translation errors in camera pose estimates, our method maintains low synchronization errors (e.g., 9.6 ms for *Fencing* and 19.4 ms for *Tagging*), indicating that **pose noise does not directly translate to sync failure**.
>
> | Sport         | Cam Rot Err ↓ | Cam Trans Err ↓ | δ median ↓ |
> |---------------|---------------|------------------|-------------------|
> | Fencing       | 10.86        | 13.96          | 9.6                  |
> | Volleyball    | 8.53         | 14.62          | 30.8               |
> | LegoAssemble  | 3.12         | 7.07          |   38.3              |
> | Badminton     | 10.24         | 11.82          | 34.6                 |
> | Tagging       | 5.82         | 9.10          | 19.4              |
> | Basketball    | 4.00         | 12.23          | 27.5                 |
> | Tennis    | 2.73         | 13.89          | 113.6                |
>
> > **Note:** High sync error in **Tennis** is due to **distant camera placement**, resulting in minimal observable motion (see supplementary video).
>
> We further demonstrate our system's effectiveness by applying our method to challenging in-the-wild videos, as shown in the supplementary material (**Supp Fig. 3 and Video**). These include NBA basketball sequences and FIFA soccer sequences with noisy estimated camera poses. Our method still achieves good synchronization accuracy in these challenging real-world scenarios.
>
> ---
>
> ### b) Computational cost and scalability
>
> We have included a runtime and resource efficiency comparison against baselines such as **MAST3R**, **Uni4D**, and **Sync-NeRF** on the **Panoptic** dataset (15 videos, ~200 frames each) using **a single A6000 GPU**.
>
> Our method is **training-free**, and most of the overhead comes from **visual cue extraction**. While we acknowledge that further optimizations (e.g., lightweight modules) can improve runtime, **efficiency is not the primary focus of our work**, which targets accuracy and robustness under challenging conditions. In our experiments, we show our system could accurate synchronize 30 cameras on the **CMU Panoptic** dataset. With additional computing resources and parallel video preprocessing, the overall running time can be significantly reduced.
>
> | Method           | Runtime (hrs)  |
> |------------------|---------------|
> | Uni4D* [58]       |       3.9        |
> | Mast3R [28]       |         1.2      |
> | Sync-NeRF* [24]   |       4.2      |
> | **Ours**          |       3.3        |
>
> In practice, our method is fast enough for typical multi-camera applications (e.g., 4-5 camera), such as movie production, sports analysis, and video surveillance, where offline processing is acceptable. We see further efficiency improvements (e.g., via lightweight models) as a valuable direction for future work, especially for very large-scale or latency-sensitive scenarios.
>
> ---
> ### c) Failure case analysis
>
> We provide detailed failure cases analysis of each module within our framework and qualitative examples in **Supp.F**. We will further improve the discussion of limitations and robustness of our method in future version.

---

> > ### Comment · Reviewer_dtE8 · 2025-08-04
> >
> > Thanks, I've read both the response and the review comments from others. Most of my concerns are being tackled. Hence, I am happy to keep my original positive score (borderline accept).

---

### Official Review · Reviewer_pwqc · 2025-07-01

**Clarity:** 3
**Significance:** 2
**Originality:** 2
**Rating:** 4
**Confidence:** 4

**Summary:**

The paper presents a novel framework for synchronizing unsynchronized multi-camera videos capturing dynamic scenes. It addresses a common challenge in everyday scenarios—such as sports events, concerts, or family gatherings—where videos from multiple handheld devices lack time alignment. By leveraging cross-view object motion, the method recovers globally consistent temporal offsets with millisecond accuracy, enabling applications like 4D scene reconstruction and novel-view synthesis.

**Questions:**

See in Weaknesses.

**Ethical Concerns:**

["NO or VERY MINOR ethics concerns only"]

**Final Justification:**

The rebuttal and discussions addressed my concerns about the paper contributions. I tend to raise the score.

**Limitations:**

Yes.

**Quality:**

2

**Strengths And Weaknesses:**

Strengths:
1. By integrating advanced models (e.g., CoTracker3 for trajectory tracking and MAST3R for cross-view matching), the method avoids training from scratch, thereby improving efficiency (Section 3.3, Stage 0).
2. The method significantly outperforms baselines on Egohumans (δ_med=46.6ms) and CMU Panoptic (δ_med=41.5ms), particularly in complex scenarios.

Weaknesses:

The core idea — that synchronized frames of dynamic points should obey epipolar constraints is simple, which seems like more of an effective integration of recent tools (e.g., VGGT, CoTracker, MAST3R).

1. The method assumes a fixed frame rate, but in real-world videos, inconsistent frame rates may limit its applicability (Limitations). It is recommended to extend the approach to a variable frame rate model.
2. It is recommended to add comparisons of resource efficiency between methods.

3. Expand the comparison to include PoseSync[59] (human pose priors) and audio synchronization[46], strengthening the validation of the 'assumption-free' approach.

4. There shall be many camera synchronization methods in the literature, including traditional and deep learning methods. The study and comparison seem not enough.

---

> ### Author Rebuttal · Authors · 2025-07-31
>
> ### a) Effective Integration of Recent Tools
>
> We agree that the core idea—synchronized frames should obey epipolar constraints—is conceptually simple. However, we see this as a **strength** rather than a limitation. Synchronizing dynamic, unconstrained videos remains a highly challenging task, and a method that is both **simple and effective** is valuable in real-world settings.
>
> While our method builds on recent advances such as **VGGT**, **CoTracker**, and **MAST3R**, our **main contribution lies in the unified energy formulation and inference strategy** that integrates both static and dynamic cues to solve a **new problem these tools alone cannot address**. We appreciate the reviewer’s recognition of our integration efforts, and believe that the clarity of our formulation is key to the method’s robustness across diverse settings.
>
> ---
> ### b) Inconsistent frame rates
> First, we would like to clarify that our method can **synchronize videos with different (but known) frame rates very well**. In practice, frame rate information is usually available from video metadata, including for internet videos. Our main limitation is with videos that have varying motion speeds within a single clip (e.g., segments with both slow-motion and fast-motion or non-uniform motion), not with differing constant frame rates across videos.
>
> To demonstrate robustness and adaptability of our method to varying frame rates, we further evaluated our method on the **CMU Panoptic** dataset underlying 2 challenging scenarios: **1) known varying frame rates** and **2) unknown varying frame rates**. Videos are sampled at frame rates of 5-30 fps. We directly apply our method **without any pipeline changes** in scenario 1). For scenario 2), we **remove the assumption of known frame rates** and follow the extension detailed in the **Supp. E**. We present experimental results along with synchronization performance on known fixed 30-fps reported in **Tab 1** for reference. Our method performs comparably under varying known FPS (median: **51.5 ms**) to the fixed 30 fps setting (median: **41.5 ms**). When frame rates are unknown, performance degrades gracefully (median: **80.5 ms**) due to additional uncertainty, demonstrating robustness of our framework even without frame rate metadata.
>
> | Method         | δ mean ↓ | δ median ↓ |
> |----------------|--------------------|--------------------|
> | Constant Known FPS        |     112.6      |     41.5       |
> | Varying Known FPS        |     103.9      |     51.5       |
> | Varying Unknown FPS        |     164.2      |     80.5       |
>
> ---
>
> ### c) Runtime comparison
>
> Thank you for the suggestion. We have included a runtime and resource efficiency comparison against baselines such as **MAST3R**, **Uni4D**, and **Sync-NeRF** on the **Panoptic** dataset (15 videos, ~200 frames each) using **a single A6000 GPU**.
>
> Our method is **training-free**, and most of the overhead comes from **visual cue extraction**. While we acknowledge that further optimizations (e.g., lightweight modules) can improve runtime, **efficiency is not the primary focus of our work**, which targets accuracy and robustness under challenging conditions. With additional computing resources and parallel video preprocessing, the overall running time can be significantly reduced. We view efficiency-focused improvements as an important direction for future work.
>
> | Method           | Runtime (hrs)  |
> |------------------|---------------|
> | Uni4D* [58]       |       3.9        |
> | Mast3R [28]       |         1.2      |
> | Sync-NeRF* [24]   |       4.2      |
> | **Ours**          |       3.3        |
>
> ### d) PoseSync and audio synchronization
>
> Thank you for the suggestion. While both PoseSync [59] and audio-based synchronization [46] are valuable efforts, they are **not directly comparable** due to differences in assumptions and problem settings.
>
> **PoseSync** assumes a **single moving human** and leverages **pose priors**, whereas our datasets feature **multiple moving people** or **non-human subjects** (e.g., pigeons in *3DPOP* and dynamic objects in sync-NeRF), making direct comparison infeasible. That said, we agree it would be useful to include PoseSync on single-human datasets, and will add such experiments in the final version (beyond the rebuttal timeline).
>
> **Audio-based methods** rely on additional audio input and make different assumptions. While orthogonal to our visual-only setup, we agree it is important to **clarify this distinction**, and we will add a more detailed discussion in the Related Work section to better position our **assumption-free, general-purpose approach**.
>
> ---
>
> ### e) Additional prior works
>
> We respectfully note that our selected baselines are **competitive and representative** for general video synchronization—an underexplored problem—spanning both **traditional geometric** and **deep learning** approaches. While we acknowledge the reviewer’s concern, the comment (“shall be many” and “seem not enough”) is a bit general to us. We would **greatly appreciate any concrete recommendations** on additional methods relevant and feasible for evaluation in our setting of **generic multi-view time synchronization**, and we will gladly consider incorporating them in our revision.
>
> In the meantime, we will expand the **Related Work** section to better articulate our baseline choices and further situate our contributions within the broader literature.

---

> > ### Author Response · Authors · 2025-08-06
> >
> > Hi, we would like to kindly follow up to ensure that your concerns are being properly addressed. Please don’t hesitate to let us know if there are any remaining questions or if additional information or experiments would be helpful. Thank you again for reviewing our paper!

---

> > ### Comment · Reviewer_pwqc · 2025-08-09
> > **Thanks for the response**
> >
> > Thanks for the authors' response and additional experiments for addressing my concerns. But I still feel the following questions need to be emphasized. It's ok, the authors don't need to answer them at this moment. Just want to think about them when reading the paper.
> >
> > 1) Epipolar line constraint certainly is effective, and has been studied and used for camera sync.
> > I notice that the comparisons are just three methods. Can we use traditional methods, such as
> > On the two-view geometry of unsynchronized cameras. In: Proceedings of the IEEE Conference on Computer
> > Vision and Pattern Recognition. pp. 4847–4856 (2017), by providing it with correspondence sets predicted with modern computer vision techniques as in the paper? What are the main advantages here?
> >
> > 2) As admitted by the authors, "our key contribution is to leverage recent advances in computer vision, specifically dense tracking, cross-view correspondences, and robust structure-from-motion, to build a robust and versatile system that can reliably synchronize challenging videos." Is the technique contribution of the paper enough or not?
> >
> > I will give the final score later.

---

> > > ### Author Response · Authors · 2025-08-09
> > >
> > > Dear reviewer,
> > >
> > > Thank you so much for your questions.  We appreciate your deeper discussion with our work and would like to address your concerns.
> > >
> > > ---
> > >
> > > ###  a) Traditional method [3]
> > >
> > > In the related work section, we discussed [3], which makes a strong assumption of **a static camera with a fixed relative pose**. In contrast, our approach can handle **dynamic cameras**, even under extremely challenging egomotions, and works robustly for in-the-wild scenarios (see Supplementary Videos). Moreover, the two methods differ significantly in both their energy formulations and underlying motivations given the difference in assumption, where the linearization in [3] doesn't hold true for moving camera scene under our setting.
> > >
> > >  ---
> > >
> > > ### b) Contributions
> > >
> > > Our pipeline incorporates state-of-the-art tools such as VGGT, MAST3R, and CoTracker; however, **our primary contribution lies in the unified energy formulation and inference strategy**, which jointly leverage static and dynamic cues to solve a **new problem that these tools alone cannot address**. Compared to existing work on camera synchronization, we tackle a far more challenging scenario in which both cameras and objects are in motion, with complex dynamics and diverse viewpoints across multiple views, rather than focusing solely on a single fixed two-view synchronization setting. Our proposed energy formulation allows us to solve synchronizations **across more than 30 cameras**.
> > >
> > > ---
> > >
> > > We hope we've addressed your concerns and welcome any further discussion.

---

### Official Review · Reviewer_5Bep · 2025-07-02

**Clarity:** 3
**Significance:** 2
**Originality:** 4
**Rating:** 4
**Confidence:** 3

**Summary:**

This paper introduces a 3D geometry-based method for synchronizing multi-view videos of the same scene that have different starting recording times. It leverages off-the-shelf techniques to acquire camera poses, cross-view correspondences, and per-view 2D point tracks. The authors develop an optimization method that uses epipolar constraints to optimize the time offset for each video. The paper evaluates its method on various multi-view datasets, demonstrating its outperformance over baseline methods. However, the proposed method has several limitations: it cannot handle videos with different FPS, assumes ideal camera poses, and its optimization can be hindered by outlier correspondence matches. Overall, despite addressing a practical problem, the proposed solution still has significant limitations.

**Questions:**

- The optimization operates on 2D point tracks at discrete timestamps. While a global offset estimation may approximate a continuous-time solution, it's unclear if this leads to sub-optimal results. If the input videos have low framerates (i.e, the point track observations become more discrete), will the optimization results significantly degrade?

- In the experiments, many videos have stationary cameras rather than moving cameras. Would the stationary camera be a better anchor for synchronization? Would the method perform worse with a set of videos where all cameras are in motion?

- The paper uses a neural radience field method (K-Planes) as a proxy to demonstrate the effectiveness of synchronization. However, K-Planes is not a state-of-the-art method. Is there any reason not to use more recent methods or explicit representation like dynamic 3D Gaussian splatting?

**Ethical Concerns:**

["NO or VERY MINOR ethics concerns only"]

**Final Justification:**

The rebuttal has addressed my concerns.

**Limitations:**

Yes.

**Quality:**

2

**Strengths And Weaknesses:**

**Strengths**
- This paper tackles a practical problem relevant to real-world multi-view video applications.
- The authors perform a thorough evaluation across diverse multi-view datasets and include comprehensive ablation studies on the algorithm's design.
- The overall presentation is clear and well-structured.

**Weaknesses**
- High sensitivity to input quality: The method's performance heavily depends on the accuracy of upstream components, specifically camera pose estimations, 2D point tracks, and cross-view correspondences. In Table 3, utilizing ideal ground truth data for poses and correspondences yields significantly higher accuracy (e.g., 94.8 vs. 40.2 in the metric AUC@100). A major limitation is the method's inability to refine or mitigate errors from imperfect estimations and outliers in these inputs, which inherently degrades the final synchronization results.
- Insufficient quantitative performance: the quantitative evaluation reveals significant synchronization errors, averaging around 100ms. This margin remains substantial (e.g., ~3 frames at 30 FPS) and raises concerns about the method's practical utility for precise synchronization.
- Limited frame rate handling: The method presented in the main paper assumes all input videos share an identical frame rate (FPS). While the supplementary material briefly demonstrates an extension for different frame rates, this crucial aspect lacks a thorough experimental evaluation.
- Minor writing issue:
  - Table 2 caption. "we show mean and median error", while this table shows AUC instead of errors.

---

> ### Author Rebuttal · Authors · 2025-07-31
>
> ### a) Sensitivity to input quality
>
> We clarify that **AUC@100** in **Tab. 3** is an intermediate metric across all video pairs, including those with opposite viewpoints or no temporal overlap. For ground-truth evaluation (**1st** row in Tab. 3), we use dataset-provided GT poses and correspondences for all pairs—regardless of overlap—to demonstrate that our system produces near-perfect output when given perfect input.
>
> In practical scenarios, the more relevant comparison is between the **bottom row** and **2nd** rows. On the challenging EgoHumans dataset, our framework achieves a median error of 46.6 ms, which is close to the 28.6 ms oracle result obtained with ground-truth poses—demonstrating robust performance with imperfect inputs.
>
> As further shown in **5th** row in **Tab. 4**, our method leverages the energy landscape to filter spurious pairs, highlighting its robustness to imperfect inputs.
>
>  ---
>
> ### b) Quantatitive performance
>
> The mean synchronization error ( \~100 ms) is heavily influenced by a small number of extremely challenging camera views, as we did not exclude any views from our evaluation. The **Supplementary Video** provides visual examples to highlight the difficulty of synchronization (e.g. synchronization between third-person and egocentric cameras with rapid motion blur). Notably, the median error is much lower (\~40 ms), indicating that our method achieves strong synchronization accuracy for the majority of cameras. We also report error percentiles on EgoHumans dataset below, showing that the higher mean is driven by a minority of extreme cases.
>
>
> | Percentile      | 25th       | 50th (Median) | 75th       | 100th (Max) |
> |-----------------|------------|----------------|------------|--------------|
> | Error (ms) ↓    | 16.3   | 46.6        | 156.9   | 1798.2   |
>
> ---
>
> ### c) Diverse frame rate handling
>
> First, we would like to clarify that our method can **synchronize videos with different (but known) frame rates very well**. In practice, frame rate information is usually available from video metadata, including for internet videos. Our main limitation is with videos that have varying motion speeds within a single clip (e.g., segments with both slow-motion and fast-motion or non-uniform motion), not with differing constant frame rates across videos.
>
> To demonstrate robustness and adaptability of our method to varying frame rates, we further evaluated our method on the **CMU Panoptic** dataset underlying 2 challenging scenarios: **1) known varying frame rates** and **2) unknown varying frame rates**. Videos are sampled at frame rates of 5-30 fps. We directly apply our method **without any pipeline changes** in scenario 1). For scenario 2), we **remove the assumption of known frame rates** and follow the extension detailed in the **Supp. E**. We present experimental results along with synchronization performance on known fixed 30-fps reported in **Tab 1** for reference. Our method performs comparably under varying known FPS (median: **51.5 ms**) to the fixed 30 fps setting (median: **41.5 ms**). When frame rates are unknown, performance degrades gracefully (median: **80.5 ms**) due to additional uncertainty, demonstrating robustness of our framework even without frame rate metadata.
>
> | Method         | δ mean ↓ | δ median ↓ |
> |----------------|--------------------|--------------------|
> | Constant Known FPS        |     112.6      |     41.5       |
> | Varying Known FPS        |     103.9      |     51.5       |
> | Varying Unknown FPS        |     164.2      |     80.5       |
>
> ---
> ### d) Optimization on low frame rates
>
> While our intermediate pairwise offset estimation operates at discrete timestamps, the **final synchronization output is continuous in time**. This design reflects the frame-based nature of input videos, point tracks, and camera pose estimates.
>
> We also experimented with **interpolating point tracks and camera poses** to enable continuous-time matching. However, we observed **no significant performance improvement**, likely because the discrete matching better couples with the observed data.
>
> To test robustness to low frame rates, we downsampled the **CMU Panoptic** dataset from original **30fps** to **15 fps**. As expected, performance slightly degrades due to reduced temporal overlap, but our method **still achieves reasonable synchronization accuracy**, demonstrating resilience under challenging conditions. We want to further add that our preprocessing methods, Co-Tracker and DEVA, are designed for video input and typically assume a high frame rate, making 15 FPS uncharacteristically low for optimal performance.
>
> | FPS         | δ mean ↓ | δ median ↓ |
> |----------------|--------------------|--------------------|
> | 30 (original)     |    112.6       |    41.5        |
> | 15     |       157.2    |      45.6      |
>
> ---
> ### e) Stationary vs. moving cameras
>
> It is not necessarily true that static cameras make better anchors. The major benefit is static cameras often offer more accurate pose estimates, but they may suffer from **limited viewpoint or temporal overlap** with others. In contrast, dynamic cameras can provide **richer visual connections**, which our method effectively exploits.
>
> As shown in our **supplementary videos**, our approach works well even when **all cameras are moving** or when **pose estimates are noisy**, including challenging in-the-wild cases.
>
> To validate this, we conduct an ablation using **only dynamic cameras** in selected *EgoHumans* sequences—specifically those without opposing viewpoints (e.g., *volleyball*, *badminton*) or with two opposing players (e.g., *tennis*, *fencing*). As shown in the below table, the fist column (static & dynamic) leverages both static and dynamics cameras for synchronization while the second column (dynamic-only) only use dynamic cameras. We report the dynamic camera synchronization performance in the table. As show below, the synchronization performance remains strong with only dynamic cameras input, confirming that our method is **effective for both static and dynamic camera setups**.
>
> | Method         | Static & Dynamic             |                      | Dynamic-only             |                      |
> |----------------|---------------------|----------------------|---------------------|----------------------|
> |                | δ mean ↓  | δ median ↓    | δ mean ↓  | δ median ↓    |
> | Legoassemble       |         45.9            |    25.0                  |           50.0       |       50.0             |
> | tagging       |     31.1                |       25.9               |               25.0      |         0.0             |
> | basketball       |    29.1                 |   30.7                   |             12.5        |              0.0        |                    |
>
> We also apply our method to in-the-wild videos (**Supp. Fig. 3 and Video**), including NBA basketball and FIFA soccer sequences where all cameras are moving. Our method still achieves good synchronization accuracy in these challenging real-world scenarios.
>
> ---
> ### f) Use of K-Planes vs. Dynamic 3D Gaussian Splatting
>
> Our method is **agnostic to the underlying radiance field representation** and can interface with a wide range of dynamic scene reconstruction models, including K-Planes, Dynamic Gaussian Splatting, or others.
>
> We chose K-Planes in our experiments for consistency with Sync-NeRF [24]. We agree that more recent methods such as **dynamic 3DGS** offer compelling benefits in speed and quality. Incorporating such representations to demonstrate downstream application is a valuable and very possible direction, and we will do this in our  revision to further show the versatility of our synchronized outputs for downstream 3D reconstruction and rendering.
>
>  ---
> ## g) Minor
>
> We will fix the writing issue in revision.

---

> > ### Author Response · Authors · 2025-08-06
> >
> > Hi, we would like to kindly follow up to ensure that your concerns are being properly addressed. Please don’t hesitate to let us know if there are any remaining questions or if additional information or experiments would be helpful. Thank you again for reviewing our paper!

---

> > ### Comment · Reviewer_5Bep · 2025-08-06
> >
> > The rebuttal satisfied my concerns, so I'm raising my rating to borderline accept.

---

### Official Review · Reviewer_QBkg · 2025-07-04

**Clarity:** 3
**Significance:** 3
**Originality:** 2
**Rating:** 4
**Confidence:** 5

**Summary:**

This paper presents a method for time-synchronizing multiple cameras by observing moving 3D points and ensuring they follow the epipolar constraints.   It achieves an error below 130ms.   It uses the following set of tools: camera parameters via VGGT, dense correspondences with CoTracker3, cross-view matches with MAST3R, and dynamic objects with DEVA.  The loss function, the Sampson error, is related to the squared Euclidean distance from a point to its corresponding epipolar line.

The key challenge is that each of these measurements is noisy, and the total loss function across all camera pairs and trajectories is highly nonlinear.  The method employs an exhaustive search of time alignment to find the optimal time alignment and calibration between two cameras.
This is followed by iteratively reweighted least squares for the global optimization.

**Questions:**

None.

The authors have listed their weaknesses.  Addressing them is beyond the scope of this paper.   However, it is unclear whether NeurIPS is the proper venue for this work.

**Ethical Concerns:**

["NO or VERY MINOR ethics concerns only"]

**Final Justification:**

Thank you for the rebuttal. It addressed my concerns.

**Limitations:**

none.

**Paper Formatting Concerns:**

none.

**Quality:**

3

**Strengths And Weaknesses:**

The algorithm is more effective than all existing strong baseline methods, including both geometric and learning-based approaches.   On the challenging Egohuman dataset, it reduced the time synchronization error by a factor of 4.

It also listed the weakness of the paper itself:
a) it depends on accurate camera poses;
b) it fails when there is rapid motion or large viewpoint changes
c) it does not handle frame rate variations.

---

> ### Author Rebuttal · Authors · 2025-07-31
>
> ### a)  Dependency on accurate camera poses
>
> While our method requires predicted camera poses as input, **it does not rely on accurate poses for every frame**. In practice, a **subset of frames with reliable pose estimates is sufficient**, thanks to our design—particularly the **energy landscape optimization** and **robust filtering of spurious pairs**. This allows the method to remain resilient to pose noise and gracefully degrade under imperfect predictions.
>
> This is empirically supported in two ways:
>
> 1. **Pose error vs. sync accuracy** (table below): Even with large rotation and translation errors (relative angular error in rotation and translation following VGGT) in camera pose estimates, our method maintains low synchronization errors (e.g., 9.6 ms for *Fencing* and 19.4 ms for *Tagging*), indicating that **pose noise does not directly translate to sync failure**.
>
> 2. **GT vs. estimated poses**: As shown in **Tab. 3 (2nd and bottom row)**, using estimated poses on the challenging *EgoHumans* dataset still yields a **median error of 46.6 ms**, close to the **28.6 ms** oracle result with ground-truth poses—demonstrating **robust performance even under imperfect pose inputs**.
>
> Further, **Tab. 4 (5th row)** shows how our method filters out unreliable matches via the energy landscape, enhancing stability.
>
> We acknowledge this limitation in our paper, but clarify that it is **not a binary failure mode**. We will make this point more explicit in the revision.
>
> | Sport         | Cam Rot Err ↓ | Cam Trans Err ↓ | δ median↓ |
> |---------------|---------------|------------------|-------------------|
> | Fencing       | 10.86          | 13.96             | 9.6               |
> | Volleyball    | 8.53           | 14.62             | 30.8              |
> | LegoAssemble  | 3.12           | 7.07              | 38.3              |
> | Badminton     | 10.24          | 11.82             | 34.6              |
> | Tagging       | 5.82           | 9.10              | 19.4              |
> | Basketball    | 4.00           | 12.23             | 27.5              |
> | Tennis        | 2.73           | 13.89             | 113.6             |
>
> > **Note:** High sync error in **Tennis** is due to **distant camera placement**, resulting in minimal observable motion (see supplementary video).
>
> ---
>
> ### b) Rapid motion or large viewpoint changes:
>
> We demonstrate real-world effectiveness by applying our method to challenging in-the-wild videos, as shown in the supplementary material (**Supp Fig. 3 and Video**). These include NBA basketball sequences and FIFA soccer sequences with rapid shooting motion, diverse backgrounds, and varying camera viewpoints.
>
> ---
>
> ### c) Frame rate variations:
>
>
> First, we would like to clarify that our method can **synchronize videos with different (but known) frame rates very well**. In practice, frame rate information is usually available from video metadata, including for internet videos. Our main limitation is with videos that have varying motion speeds within a single clip (e.g., segments with both slow-motion and fast-motion or non-uniform motion), not with differing constant frame rates across videos.
>
> To demonstrate robustness and adaptability of our method to varying frame rates, we further evaluated our method on the **CMU Panoptic** dataset underlying 2 challenging scenarios: **1) known varying frame rates** and **2) unknown varying frame rates**. Videos are sampled at frame rates of 5-30 fps. We directly apply our method **without any pipeline changes** in scenario 1). For scenario 2), we **remove the assumption of known frame rates** and follow the extension detailed in the **Supp. E**. We present experimental results along with synchronization performance on known fixed 30-fps reported in **Tab 1** for reference. Our method performs comparably under varying known FPS (median: **51.5 ms**) to the fixed 30 fps setting (median: **41.5 ms**). When frame rates are unknown, performance degrades gracefully (median: **80.5 ms**) due to additional uncertainty, demonstrating robustness of our framework even without frame rate metadata.
>
> | Method         | δ mean ↓ | δ median ↓ |
> |----------------|--------------------|--------------------|
> | Constant Known FPS        |     112.6      |     41.5       |
> | Varying Known FPS        |     103.9      |     51.5       |
> | Varying Unknown FPS        |     164.2      |     80.5       |
>
> ---
>
> ### d) Venue suitability:
>
> We believe our work is well-suited for NeurIPS, as it addresses a fundamental problem at the intersection of computer vision and machine learning, using tools from **visual foundation models** and **optimization-based inference**. Our approach combines insights from geometry, learning, and large-scale visual understanding, with clear implications for **downstream applications** such as 3D reconstruction and video understanding.

---

> > ### Author Response · Authors · 2025-08-06
> >
> > Hi, we would like to kindly follow up to ensure that your concerns are being properly addressed. Please don’t hesitate to let us know if there are any remaining questions or if additional information or experiments would be helpful. Thank you again for reviewing our paper!

---

### Note · Authors · 2025-08-15

Dear Reviewers and ACs,

We sincerely thank all reviewers for their constructive feedback, and greatly appreciate the AC’s efforts in facilitating discussion during the rebuttal period.

We summarize our paper’s primary contributions as below:


a) **Generic Synchronization Setting**


Compared to prior view synchronization works discussed in Related Work, we tackle a more **challenging and general** setting: multiple moving people or non-human subjects without human-pose priors [59] or single-object assumptions [29], and diverse moving cameras without fixed-view constraints [3]. Our method could also handle synchronization across 30+ cameras without excluding any extreme viewpoints, where non prior work achieve this. The supplementary video illustrates the complexity and generality of our setting.




b) **Robust Energy Formulation**


Starting from the simple observation that synchronized frames should satisfy epipolar constraints, we propose a **unified and robust** energy formulation that solves synchronization via energy minimization, applicable to all moving scenes with both object and camera motion. The resulting energy landscape could further filter out spurious pairs and remains robust to imperfect inputs.





c) **Flexible Inference Framework**

We propose a novel optimization solution to the highly non-convex energy minimization problem by leveraging both static and dynamic cues of the scene. Our framework is highly flexible, which could be directly applied to varying input video frame rates and even perform without the assumption of known frame rates, **without any pipeline changes**, as demonstrated in the rebuttal.





d) **Strong In-the-Wild Performance**


Our method shows strong performance on challenging in-the-wild videos, including cases with extreme egomotion and rapid camera movement, as shown in the supplementary video. Quantitatively, it significantly outperforms all competing baselines, particularly under complex scenes.





e) **Open Resources**


We commit to release all our codes and data to the public. We will also incorporate all suggested changes from the reviewers into the revised version.


Sincerely,

Visual Sync authors

---

### Decision · Program_Chairs · 2025-09-17

**Decision:**

Accept (poster)

**Comment:**

This paper presents Visual Sync, a training-free framework for synchronizing multi-camera videos using epipolar constraints and modern vision foundation models. The approach is novel in its unified energy formulation and shows strong empirical performance across several datasets, often surpassing prior synchronization methods. Reviewers valued the motivation, broad applicability, and solid results. Concerns remain about heavy reliance on external pretrained models, limited scalability due to O(N²) matching, and modest technical novelty beyond integration. Despite these limitations, the consensus is positive. Therefore, I recommend acceptance.